# Genome-Wide Identification, Characterization and Expression Patterns of the DBB Transcription Factor Family Genes in Wheat

**DOI:** 10.3390/ijms252111654

**Published:** 2024-10-30

**Authors:** Yalin Wang, Huimin Qin, Jinlan Ni, Tingzhi Yang, Xinru Lv, Kangzhen Ren, Xinyi Xu, Chuangyi Yang, Xuehuan Dai, Jianbin Zeng, Wenxing Liu, Dengan Xu, Wujun Ma

**Affiliations:** 1College of Agronomy, Qingdao Agricultural University, Qingdao 266109, China; wyl05260@163.com (Y.W.); hmhm02232022@163.com (H.Q.); jinlanni@163.com (J.N.); yangtingzhi3@163.com (T.Y.); lvxinru2022@163.com (X.L.); renkangzhen1030@163.com (K.R.); 19861339778@163.com (X.X.); 19560771816@163.com (C.Y.); xiaohuan514@163.com (X.D.); jianbin_zeng@qau.edu.cn (J.Z.); liuwx@qau.edu.cn (W.L.); 2School of Agriculture, Murdoch University, Perth, WA 4350, Australia

**Keywords:** phylogenetic relationships, *TaDBB*s, transcription factors, wheat

## Abstract

Double B-box (DBB) proteins are plant-specific transcription factors (TFs) that play crucial roles in plant growth and stress responses. This study investigated the classification, structure, conserved motifs, chromosomal locations, cis-elements, duplication events, expression levels, and protein interaction network of the DBB TF family genes in common wheat (*Triticum aestivum* L.). In all, twenty-seven wheat DBB genes (*TaDBB*s) with two conserved B-box domains were identified and classified into six subgroups based on sequence features. A collinearity analysis of the DBB family genes among wheat, *Arabidopsis*, and rice revealed some duplicated gene pairs and highly conserved genes in wheat. An expression pattern analysis indicated that wheat *TaDBB*s were involved in plant growth, responses to drought stress, light/dark, and abscisic acid treatment. A large number of cis-acting regulatory elements related to light response are enriched in the predicted promoter regions of 27 *TaDBB*s. Furthermore, some of TaDBBs can interact with COP1 or HY5 based on the STRING database prediction and yeast two-hybrid (Y2H) assay, indicating the potential key roles of *TaDBB*s in the light signaling pathway. Conclusively, our study revealed the potential functions and regulatory mechanisms of *TaDBB*s in plant growth and development under drought stress, light, and abscisic acid.

## 1. Introduction

Transcription factors (TFs) are involved in gene expression regulation. The zinc finger proteins, an essential group of TFs in eukaryotes, include proteins with B-box motifs, known as BBX. Within the BBX family proteins, members lacking the CCT domain at C-terminus that have at least two B-box domains are designated as Double B-box (DBB) proteins. In response to abiotic stress, *Arabidopsis* and rice BBX proteins mediate protein–protein interactions through their B-box domains, regulating the UV-B response of photomorphogenesis by interacting with CONSTITUTIVE PHOTOMORPHOGENIC 1 (COP1) and ELONGATED HYPOCOTYL 5 (HY5) [1]. Asp residues in the B-box domain are essential for transcriptional activation and DNA binding [2,3,4,5]. Yeast two-hybrid screening revealed that OsARID3 and OsPURα proteins interacted with the B-box domain of OsBBX8 under drought stress [6]. In addition, *AtBBX22* was regulated by abscisic acid (ABA) [7]. Members of the DBB subfamily exhibit diverse functions across different organs. As a zinc finger TF family, plant DBB TFs are involved in regulating numerous life activities, including seed germination [8,9], flowering [10], responses to biotic or abiotic stresses [11], and plant hormone signal transduction [12].

In model plants like *Arabidopsis* and rice, the *DBB* gene family has been extensively studied. The *Arabidopsis DBB* gene family consists of eight members [13], and most of them have been found to be related to light signal transduction and regulated by diurnal rhythms. *DBB1a* (*At2G21320*), *DBB1b* (*At4G38960*), *DBB2* (*At4G39070*), *DBB3* (*At1G78600*), *STO* (*At1G06040*), and *STH* (*At2G31380*) regulate the expression of key light signaling-related genes, including *HY5*, *COP1*, *CHS*, and key circadian clock genes *CCA1*, *LHY*, *ELF3*, and *TOC1*, playing vital roles in light-mediated plant growth [14,15,16,17,18,19]. DBB3 and STH2, interacting with HY5 and COP1, regulate early seedling development, early chloroplast formation, and anthocyanin accumulation [15,20]. Overexpression of *DBB1b* results in abscisic acid (ABA) hypersensitivity, indicating that DBB proteins may be involved in plant hormone signal transduction [21]. In chrysanthemum, CmBBX24 improves cold and drought tolerance [22]. Other genes, like *OsDBB3b* (*LOC_Os09g35880*), also interact with HY5, and it shows a conversed function compared to its homologous gene in *Arabidopsis* [14,23]. Drought stress seriously affects crop productivity. In *Arabidopsis*, transcriptional regulation of drought-inducible gene expression occurs via ABA-independent and ABA-dependent signaling pathways [24]. In rice, an ABA-dependent signaling pathway is preferred in response to drought stress [25]. So far, four *BBX* genes in *Arabidopsis* and rice, namely *AtBBX1*, *AtBBX29*, *OsBBX8*, and *OsBBX4* [26,27], have been functionally identified as drought-responsive genes. *AtBBX1* is a well-characterized flowering time regulator [28]. Furthermore, *AtBBX24*, *AtBBX5*, and *OsBBX4* [29,30] have been studied related to salinity stress tolerance. *Arabidopsis* plants that overexpressed *AtBBX24* were salinity tolerant and showed an increased root length [31]. Currently, the research on the *DBB* gene family mainly focuses on model plants such as *Arabidopsis* and rice, and the research on wheat is relatively limited.

This study identified and classified wheat DBB genes (*TaDBB*s), followed by phylogenetic analysis. The structures and conserved domains of these genes and their distribution in wheat genome were analyzed. The cis-acting elements and expression patterns of the *TaDBB* family were studied. We also investigated their expression patterns under ABA, light/dark, and drought treatments to reveal their roles in response to environmental stresses. Finally, yeast two-hybrid experiments were conducted to validate the predicted interactions between TaDBBs and COP1 and HY5.

## 2. Results

### 2.1. Identification of Wheat DBB Gene Family

To identify all of the DBB genes in the wheat genome, the Blastp program was initially used to search the wheat protein database based on 18 DBB protein sequences (Appendix A) of *Arabidopsis thaliana* and rice, followed by using the NCBI-CDD and Pfam databases to confirm the domains. A total of 27 *TaDBB* genes were identified in the wheat genome (*TaDBB1*–*27*, Appendix A). Multiple sequence alignments on the conserved B-box1 and B-box2 domains were performed among the 27 *TaDBB*s (Figure 1). Both B-box1 and B-box2 consist of 38 amino acids, with an interdomain length ranging from 8 to 22. In addition, characteristics of the *TaDBB* genes were listed in Appendix A, including their gene ID, chromosome location, coding sequence length, protein length, molecular weight (MW), isoelectric point (pI), instability index (II), aliphatic index (AI), grand average of hydropathy (GRAVY), and predicted location. The number of amino acids in TaDBB proteins ranged from 211 (TaDBB20) to 363 (TaDBB6), and the molecular weights ranged from 21,809.84 Da (TaDBB17) to 38,772.21 Da (TaDBB6). Their pI ranged from 4.72 (TaDBB13) to 6.04 (TaDBB7), indicating that all TaDBBs were acidic proteins (pI < 7.5). The II of TaDBB proteins ranged from 41.21 (TaDBB27) to 69.84 (TaDBB2), indicating that they were all unstable (II > 40). The AI ranged from 57.77 (TaDBB19) to 77.86 (TaDBB23). GRAVY values ranged from −0.634 (TaDBB19) to −0.188 (TaDBB23), with all of the values lower than 0, indicating that they were all hydrophilic. Subcellular localization predictions revealed that TaDBBs were all located in the nucleus.

### 2.2. Phylogenetic Analysis of TaDBBs

A phylogenetic analysis was performed to compare *DBB* genes from wheat, *Arabidopsis thaliana*, and *Oryza sativa* (Figure 2). Results revealed that the twenty-seven DBB proteins in wheat were classified into six distinct groups (Group I–VI), which contained nine, two, three, three, three, and seven members, respectively. Each group consisted of *DBB* genes from *Arabidopsis thaliana* or rice, indicating that the sequence and function of *DBB* were conserved among different species.

### 2.3. Conserved Motif and Gene Structure Analysis

Based on the gene sequences of *TaDBBs*, the gene structures of 27 *TaDBB* were analyzed (Figure 3). The analysis showed that the exon length and number were different among subgroups. Genes of groups I, II, IV, and VI usually contain two introns, while members of group III contain four or five introns, and members of group V contain one intron. This diversity of gene structure may affect splicing and expression, leading to different roles of TaDBBs in various biological processes. To gain a deeper understanding of the structural characteristics of wheat DBB proteins, the InterPro program was used to annotate the conserved motifs in these proteins (Appendix A). The analysis showed that all of the wheat DBB proteins contained the following two key domains: B-box1 and B-box2. The B-box1 domain corresponded to motif one, while the B-box2 domain corresponded to motif two. Both motifs were present in all TaDBBs. The other eight motifs had no information in the InterPro databases.

### 2.4. Chromosomal Localization and Collinearity Analysis

The wheat *DBB* genes were unevenly distributed across 14 out of 21 chromosomes (Figure 4). Chromosomes 6A, 6B, 6D, and 2A each contained three *DBB* genes, while chromosomes 7A, 7B, 7D, 2D, and 2B each contained two, and chromosomes 3A, 3B, 5A, 5B, and 5D each contained one. To evaluate the mechanism of *TaDBB* gene family expansion in wheat, syntenic analyses were conducted for these 27 *TaDBBs* (Figure 5, Appendix A). Segmental duplications were identified as the main drivers of *TaDBB* gene family expansion, with duplication events occurring primarily within groups. For example, in Group I, *TaDBB7* on 6B is homologous to *TaDBB1* (2D), *TaDBB2* (2A), *TaDBB3* (2B), and *TaDBB9* (6A). A homology analysis of *Arabidopsis*, rice, and wheat (Figure 6 and Appendix A) revealed that the homology was good in rice, but no homologous gene pairs were found in *Arabidopsis thaliana*.

To evaluate the evolution relationship of the gene family, Ks, Ka values, and the Ka/Ks ratio, along with the divergence time for each DBB gene pair, were assessed. The Ka/Ks ratio values for the segmental duplicated *TaDBB* gene pairs were all <1, suggesting that purifying selection had played a significant role in the evolution of the *TaDBB* gene family, with the divergence times occurring between 2.00 Mya and 29.36 Mya (Appendix A).

### 2.5. Analysis of Cis-Acting Elements in TaDBBs Promoter

A cis-acting element analysis of the 2 Kb promoter regions upstream of the translation initiation codon showed that the most abundant cis-elements in the promoter of the 27 DBB genes were light-responsive elements, with an average of 10.92 elements per gene (Figure 7, Appendix A). They were followed by ABA-responsive elements, with an average of 4.48 elements per gene. Additionally, all 27 *TaDBB* genes contained anaerobic-responsive elements, 17 genes contained drought-responsive elements, and 19 genes contained low-temperature-responsive elements. These cis-elements on the promotor of 27 *TaDBBs* further supported the diverse roles of DBB genes in responding to environmental stresses. Moreover, promoter sequences of the DBB gene family also contained elements involved in meristem expression, endosperm expression, circadian regulation, and cell cycle regulation. The presence of these elements suggested the extensive functions of the DBB gene family in plant growth and development.

### 2.6. Expression Patterns of Wheat DBB Genes

The expression data for the 27 *TaDBB* genes from the WheatOmics 1.0 database were listed in Figure 8A and Appendix A. *TaDBB4*, *TaDBB5*, *TaDBB6*, *TaDBB15*, *TaDBB18*, *TaDBB19*, *TaDBB20*, and *TaDBB21* were highly expressed in various tissues, with the highest expression levels in roots. The remaining 19 *TaDBBs* were only expressed at low levels in all tissues.

The analysis of cis-regulatory elements (CREs) in the promoter regions of *TaDBBs* suggested that these genes play roles in responding to various abiotic stresses and phytohormone treatments. Consequently, the expression patterns of the 27 *TaDBBs* under ABA treatments, drought, and light/dark were examined (Figure 8B–D, Appendix A).

For ABA treatment, eight genes (*TaDBB4*, *TaDBB5*, *TaDBB6*, *TaDBB15*, *TaDBB17*, *TaDBB21*, *TaDBB23*, and *TaDBB24*) initially increased in expression, peaking at 6 h before gradually decreasing. In contrast, 11 genes (*TaDBB2*, *TaDBB10*, *TaDBB11*, *TaDBB12*, *TaDBB14*, *TaDBB18*, *TaDBB19*, *TaDBB20*, *TaDBB25*, *TaDBB26*, and *TaDBB27*) showed initial down-regulation, reaching their lowest levels at 12 h, and then gradually increased. The other eight genes (*TaDBB1*, *TaDBB3*, *TaDBB7*, *TaDBB8*, *TaDBB9*, *TaDBB13*, *TaDBB16*, and *TaDBB22*) remained relatively unchanged during the first 24 h, but increased at 48 h (Figure 8B, Appendix A).

Under drought conditions, the expression levels of 11 *TaDBB* genes (*TaDBB1*, *TaDBB2*, *TaDBB3*, *TaDBB4*, *TaDBB5*, *TaDBB6*, *TaDBB14*, *TaDBB15*, *TaDBB16*, *TaDBB17*, *and TaDBB18*) increased firstly, peaking at 6 h when the plants responded to water scarcity. As the drought stress was relieved by re-watering, the expression levels of these genes declined, indicating a shift from stress response to recovery. Meanwhile, 11 genes (*TaDBB7*, *TaDBB8*, *TaDBB9*, *TaDBB10*, *TaDBB1*2, *TaDBB20*, *TaDBB22*, *TaDBB24*, *TaDBB25*, *TaDBB26*, and *TaDBB27*) exhibited a gradual decrease in expression (Figure 8C, Appendix A).

Under light/dark conditions, several *TaDBBs* showed significant changes in expression. Specifically, when turned to dark conditions, *TaDBB1*, *TaDBB7*, *TaDBB13*, *TaDBB16*, and *TaDBB22* were up-regulated, while *TaDBB2*, *TaDBB10*, *TaDBB11*, *TaDBB12*, *TaDBB14*, *TaDBB25*, *TaDBB26*, and *TaDBB27* were down-regulated. Notably, the expression levels of *TaDBB10*, *TaDBB11*, and *TaDBB12* under light conditions were more than ten times higher than under dark conditions (Figure 8D, Appendix A). The remaining genes were relatively stable, fluctuating less than five-folds. These differential expression patterns suggested that certain *TaDBBs* were closely linked to photoreceptive pathways, playing critical roles in regulating physiological responses to light exposure.

These expression patterns highlighted the dynamic response of *TaDBB* genes to photoperiod, drought stress, and ABA treatment, suggesting their potential roles in the environmental adaptation of wheat.

Using the STRING database, 19 proteins were predicted for interacting with TaDBB proteins (Figure 9, Appendix A). Among them, TaCOP1 and TaHY5, encoded by TraesCS6B02G356400 and TreasCS6A02G175800, respectively, were predicted to interact with TaDBBs. Thus, yeast two-hybrid (Y2H) assays were performed to validate the interaction between the TaDBB proteins and TaCOP1, as well as TaHY5. The Y2H assay results showed that TaCOP1 and TaHY5 did not exhibit self-activation. Subsequent verification of the interactions revealed that TaDBB6, TaDBB7, TaDBB11, TaDBB12, TaDBB13, TaDBB14, TaDBB16, TaDBB18, TaDBB21, TaDBB23, TaDBB25, TaDBB26, and TaDBB27 could interact with HY5; TaDBB7, TaDBB10, TaDBB12, TaDBB14, TaDBB20, TaDBB23, TaDBB26, and TaDBB27 could interact with COP1; and TaDBB7, TaDBB12, TaDBB14, TaDBB23, TaDBB26, and TaDBB27 could interact with both COP1 and HY5 (Figure 10).

## 3. Discussion

Several studies have demonstrated that the DBB TF family, a subfamily of the B-box family, plays a crucial role in regulating circadian rhythms and early photomorphogenesis in various plants, such as *Arabidopsis* [13], rice [32], maize [33], poplar [34], pepper [35,36], cotton [37], and tomato [38]. However, few studies have been conducted on wheat. Here, we performed a comprehensive analysis of the 27 *TaDBB* members in wheat, including their phylogenetic relationships, conserved motifs, gene structure, chromosomal positions, expression profile, and protein interaction network.

### 3.1. TaDBB Gene Family in Wheat

A phylogenetic analysis was performed to compare *DBB* genes from wheat, Arabidopsis, and rice (Figure 2). Results indicated that the sequence and function of *DBBs* were conserved among different species. However, *DBBs* in wheat have a closer phylogenetic relationship and better collinearity with rice than *Arabidopsis thaliana* (Figure 6). Moreover, the intron/exon structure of the 27 *TaDBBs* has differences, but it shares common characteristics in the same phylogenetic branch (Figure 3). Thus, the conservation and diversity of motif and intron/exon structures contribute to studying the evolution of gene families.

Although the main drivers of gene family expansion include genome polyploidy and duplication, in the wheat *DBB* gene family, 27 fragment gene pairs were detected, but no tandem gene pairs were found, indicating that genome polyploidy was the main driver of the expansion of the wheat *DBB* gene family. The divergence time ranged from 2.00 to 29.33 Mya, and most Ka/Ks are <1, indicating that members of the *TaDBB* gene family have undergone a strong purifying selection pressure during evolution.

### 3.2. Potential Roles of TaDBB Family in Wheat

Abiotic stresses, such as drought, hormonal changes, high temperatures, and nutrient deficiencies, activate genes essential for stress resistance [22]. Our study identified numerous stress-responsive CREs in the promoter regions of the 27 *TaDBB* genes. Notably, all of these genes contain light-responsive CREs, indicating their likely roles in light-mediated regulatory pathways. Similar responses have also been observed in other plants, such as maize [33] and poplar [34]. qRT-PCR results showed that *TaDBB*s were differentially expressed in light/dark environments (Figure 8, Appendix A). The *DBB* family has been shown to be sensitive to ABA [39,40]. Overexpression of *AtBBX24* in *Arabidopsis* improves osmotic stress and cold resistance, and increases sensitivity to ABA [39]. All identified *TaDBB*s, except for *TaDBB23*, were found to have promoters containing ABA elements, supporting their important roles in the ABA signaling pathway. qRT-PCR results showed that the *TaDBBs* have a different expression pattern after ABA treatment. Subfamilies II and VI initially increased and then decreased after ABA treatment, while subfamilies III and IV initially decreased and then increased. Subfamilies I and V were relatively stable at the beginning and increased after 48 h of ABA treatment. The subfamilies exhibited distinct responses to ABA treatment, suggesting that they play roles at different stages of the ABA treatment process. Therefore, these *TaDBB* genes could participate in plant hormone signaling pathways and regulate plant responses to abiotic stress.

*DBB* TFs participate in plant growth and development, participating in processes such as seedling photomorphogenesis, flowering time, phytochrome signaling, pigment deposition, and cotyledon development in species such as *Arabidopsis*, rice, maize, cotton, and tomato [23,32,33,34,35,36,37]. In this study, the expression levels of twenty-seven *TaDBBs* were detected in five different tissues at various developmental stages based on previously reported transcriptome data. The results suggest that these *TaDBB*s play roles in regulating plant growth and development, with certain *TaDBBs* potentially having unique functions in specific tissues and developmental stages. For example, *TaDBB4*, *TaDBB5*, *TaDBB6*, *TaDBB15*, *TaDBB18*, *TaDBB19*, *TaDBB20*, and *TaDBB21* are highly expressed in different tissues of wheat, suggesting that it may be involved in the basic physiological functions of wheat and extensive regulatory networks, significantly affecting the overall growth and environmental adaptability of wheat. In *Arabidopsis*, *At4G39070* is involved in drought and salt stress [41], and *TaDBB7*, *TaDBB8*, and *TaDBB9* are homologous genes of *At4G39070* that are specifically expressed in roots, indicating that they may be related to drought and salt stress in wheat.

### 3.3. Potential Roles of TaCOP1-TaHY5-TaDBB Module in Wheat

Our study revealed that light-responsive CREs are the most redundant in the promoter regions of the 27 *TaDBB* genes, and protein interaction prediction showed that 19 proteins interacted with the 27 TaDBB proteins, including TaCOP1 and TaHY5. It is well-established that the COP1–HY5–BBX module is a key player in mediating light signaling in plants [42,43,44,45]. Thus, we validated the interaction between the TaDBBs and HY5, as well as COP1, using Y2H assays. Y2H assay results showed that TaDBB7, TaDBB12, TaDBB14, TaDBB23, TaDBB26, and TaDBB27 could interact with COP1 and HY5, further highlighting the role of these TFs in light signal transduction. Combining public transcriptome data and qRT-PCR data analysis, we found that TaDBB14, TaDBB26, and TaDBB27 had large differences in expression in light/darkness, and their expression levels in roots were significantly lower than those in other parts (Figure 8). These indicate that the three proteins may be involved in regulating key processes such as photomorphogenesis and responses to changes in environmental light. The involvement of multiple TaDBB proteins in this module highlights their potential significance in the broader regulatory networks that control plant development and adaptation to light conditions.

## 4. Materials and Methods

### 4.1. Identification and Classification of TaDBBs

To investigate the *DBB* gene family in wheat, we obtained the complete genome, protein sequence, and annotation from the Ensembl Plants database (https://plants.ensembl.org/index.html, accessed on 22 August 2024) [31]. The genes we analyzed are as follows: *Arabidopsis thaliana* (*At4G38960.3*, *At2G21320.1*, *At4G39070.1*, *At4G10240.1*, *At1G78600.2*, *At1G75540.1*, *At2G31380.1*) and rice (*LOC_Os05g11510.1*, *LOC_Os01g10580.1*, *LOC_Os9g35880.1*, *LOC_Os04g41560.2*, *LOC_Os02g39360.1*, *LOC_Os06g49880.1*, *LOC_Os04g45690.1*, *LOC_Os02g43170.1*, *LOC_Os12g10660.1*, *LOC_Os06g05890.1*). DBB family members are obtained from PlantTFDB (https://planttfdb.gao-lab.org/, accessed on 22 August 2024). The corresponding sequences of *Arabidopsis thaliana* and rice were obtained from the TAIR (https://www.arabidopsis.org/, accessed on 22 August 2024) and RGAP databases (http://rice.uga.edu/, accessed on 22 August 2024), respectively. Using these sequences as queries, we performed BLASTP searches against the wheat proteins with a threshold of *E*-value < 1 × 10^−5^ to identify homologs. Further, we downloaded the Hidden Markov Model (HMM) file for DBB proteins (PF00643) from Browse-InterPro (https://www.ebi.ac.uk/interpro/, accessed on 23 August 2024) and applied it to our filtered wheat DBB protein sequences using the HMMER3.0 software to ensure comprehensive identification of family members. After removing redundancies, we compiled the final set of candidate DBB proteins. We then predicted the physicochemical properties of these proteins, including amino acid composition, isoelectric point, and molecular weight, using the ExPASy tool (https://web.expasy.org/compute_pi/, accessed on 23 August 2024), and assessed their subcellular localization via the Plant-mPLoc tool (http://www.csbio.sjtu.edu.cn/bioinf/plant-multi/#, accessed on 23 August 2024).

### 4.2. Phylogenetic Analysis of the Wheat DBB Gene Family

Multiple alignments of the conserved TaDBB protein sequences were performed using the ClustalW tool. The maximum-likelihood (ML) phylogenetic tree was constructed based on the conserved blocks and using the best-fit model with MEGA 11 software with 1000 bootstrap replications [46]. The phylogenetic tree was visualized using IToL (https://itol.embl.de/upload.cgi, accessed on 25 August 2024). 

The gene structure of the wheat DBB genes was illustrated using TBtools V2.136 software, and the conserved motifs of the wheat DBB gene family proteins were predicted using the MEME tool (https://meme-suite.org/meme/tools/meme, accessed on 25 August 2023), with the number of motifs set to 10. The evolutionary tree, the obtained motifs, and the wheat annotation file GFF3 were placed in the Gene Structure View (Advance) tool of TBtools to visualize the motifs and gene structures of the DBB family members [47].

### 4.3. Analysis of Chromosomal Localization and Gene Collinearity

The gff3 files of *Triticum aestivum* (IWGSC RefSeq 1.1), Oryza sativa Japonica Group (RGAP7), and *Arabidopsis thaliana* (TAIR10) were downloaded from the Ensembl Plants database. We then used the Gene Location Visualize from the GTF/GFF tool of TBtools to draw the physical map of DBB family members on chromosomes based on the annotation information in the GFF3 gene annotation file.

The genome sequences and annotation information of these wheat genes were obtained from the Ensembl Plants database for colinearity analysis. Fasta Stats in TBtools was used to obtain gene positions. Table Row in TBtools was used to calculate gene density. The Advanced Circos tool in TBtools was used to draw colinearity diagrams of DBB family members within species, and to adjust relevant parameters. The Dual Systeny Plot for the McscanX tool in TBtools was used to draw colinearity diagrams of DBB family members with *Arabidopsis* and rice [47,48]. The Ka/Ks ratio was calculated using the Simple Ka/Ks Calculator (NG), and the divergence time of collinear gene pairs was estimated as Ks/(2 × 9.1 × 10^−9^), with 9.1 × 10^−9^ being the mutation rate per base, per year [49].

### 4.4. Analysis of Cis-Acting Elements in Wheat DBB Genes

The PlantCARE tool (http://bioinformatics.psb.ugent.be/webtools/plantcare/html/, accessed on 12 September 2024) was used to analyze the cis-acting elements in the 2000 bp promoter region upstream of the genes’ translation start codon, and the results were visualized using TBtools software.

### 4.5. Transcriptome Analysis of TaDBBs in Different Tissues

The “Hexaploid Wheat Expression Database” in the Wheatomics 1.0 database (http://wheatomics.sdau.edu.cn/, accessed on 1 September 2024) was used. The “Chinese Spring Development (single) [50]” was selected to download the wheat leaf, stem, root, grain, and ear tissue expression data in the format of transcripts per kilobase per million mapped reads (TPM). The log2 (TPM + 1) values of the 27 *TaDBB* genes were then used to construct an expression heat map and plotted using TBtools.

### 4.6. TaDBB Expression Profiling and qRT-PCR Analysis

In this study, the effects of ABA treatment, drought stress, and light/dark conditions on the 27 *TaDBB* genes’ expression in Chinese Spring wheat seedlings were studied. Initially, seeds were sterilized, germinated on wet filter paper, and then transferred to nutrient soil. Seedlings were grown under controlled conditions with a 16-h light/8-h dark cycle at temperatures of 23 °C during the day and 18 °C at night.

For drought stress treatment, seedlings were withheld from watering during the Feeks1 to Feeks2 stage, and seedlings at the Feeks2 stage were watered again to simulate drought recovery. ABA treatment was performed using seedlings during the Feeks2 stage, and 100 μM of ABA was evenly spread to the leaves at a rate of 1 mL per plant. Leaf samples were systematically collected at specific time points after treatment (i.e., 0, 1, 3, 6, 12, 24, and 48 h). For light/dark experiments, seedlings were transferred to constant darkness during the Feeks2 stage to assess their response to light deprivation, and leaf samples were systematically collected 48 h after treatment. The same rigorous collection procedures were followed, including time point sampling, quick freezing in liquid nitrogen, and storage at −80 °C for subsequent RNA extraction. All trials were performed with three biological replicates.

The leaf samples were processed to extract the total RNA with the RNA Prep Pure Plant Plus Kit (TIANGEN., Qingdao, China), and then the HiScript III RT SuperMix for qPCR (Vazyme., Nanjing, China) was used to prepare cDNA. Primers were designed based on *TaDBB* sequences using primer 3.0 online. The specificity of each pair of primers was checked by dissociation curve analysis. The reverse-transcribed cDNA (20 μL) was diluted to a final volume of 400 μL and used for quantitative PCR. Additionally, the ChamQ Universal SYBR qPCR Master Mix (Vazyme., Nanjing, China) was employed for qPCR analyses conducted with QuantStudio™ 3 Real-Time PCR Instruments (Applied Biosystems, Thermo Fisher Scientific, Waltham, MA, USA). The qPCR procedure is 95 °C for 30 s, 40 cycles of 95 °C for 10 s, and 60 °C for 30 s. Each reaction contained 10 μL of SYBR qPCR Master Mix, 2 μL of cDNA samples, and 1 μL of each primer (10 μM) in a reaction system of 20 μL. A melt curve analysis was conducted from 60 to 95 °C with 0.5 °C increments (5 s per increment). The primers are listed in Appendix A. *GAPDH* was selected as internal references for the gene expression analyses, with the 2^−∆∆Ct^ method being employed to measure the relative expression. Gene expression data were used for analysis after log2 transformed. Gene expression at 0 h and light treatment were set as controls, and heatmaps were drawn using TBtools.

### 4.7. Prediction and Validation of TaDBBs Protein Interaction Networks

Protein interaction networks of the *TaDBB* gene family were predicted using the STRING database and visualized using Cytoscape V3.7.2 software.

The different parts of Chinese Spring wheat (leaves, seeds, roots, and stems) were processed to extract the total RNA using the RNA prep pure plant Plus Kit (TIANGEN., Qingdao, China), after which the PrimeScript™ II 1st Strand cDNA Synthesis Kit (Takara, Beijing, China) was used to prepare the cDNA. The sequences of *TaDBBs*, *COP1*, and *HY5* were obtained by nested PCR, using the primers listed in Appendix A. The 27 *TaDBBs* CDS sequences (Appendix A) were inserted into the pGADT7 vector alone. The complete CDS sequences (Appendix A) of *COP1* and *HY5* were inserted into the pGBKT7 vector, respectively. Self-activation assays of COP1 and HY5 were co-transfected with a pair of empty pGADT7 and pGBKT7 (COP1 or HY5) plasmid into Y2HGold. A total of 54 pairs of pGBKT7 (*COP1* or *HY5*) and pGADT7 (27 *TaDBBs*) plasmids were co-transformed into the yeast strain Y2HGold, followed by a selection of positive transformants on synthetic defined (SD) medium lacking Leu and Trp (SD/−Trp/−Leu). Positive colonies were then grown in a SD/−Leu/−Trp medium, collected by centrifugation, and resuspended in H_2_O before being spotted onto SD/−Trp/−Leu/ and SD/−Leu/−Trp/−His/−Ade plates. The plates were incubated at 30 °C for 3–5 d.

## 5. Conclusions

This study comprised a comprehensive study of the twenty-seven members of the wheat DBB family (*TaDBB1*–*TaDBB27*), which were categorized into six groups. As a result, these *TaDBB* genes underwent several segmental duplication events, which played a dominant role in the expansion of this gene family. These genes were also subjected to transcriptome analysis and analysis of qRT-PCR data, cis-acting elements, regulatory network analysis, and interaction analysis with COP1 and HY5. These results lay a foundation for future studies to characterize wheat *DBB* genes.

## Figures and Tables

**Figure 1 ijms-25-11654-f001:**
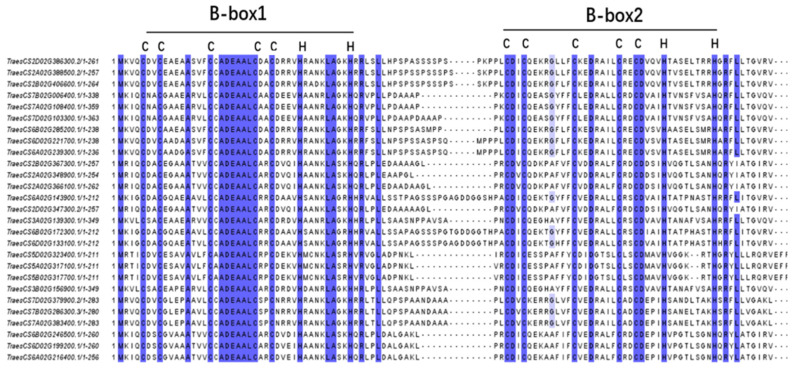
Multiple sequence alignment of conserved domains in 27 TaDBB proteins.

**Figure 2 ijms-25-11654-f002:**
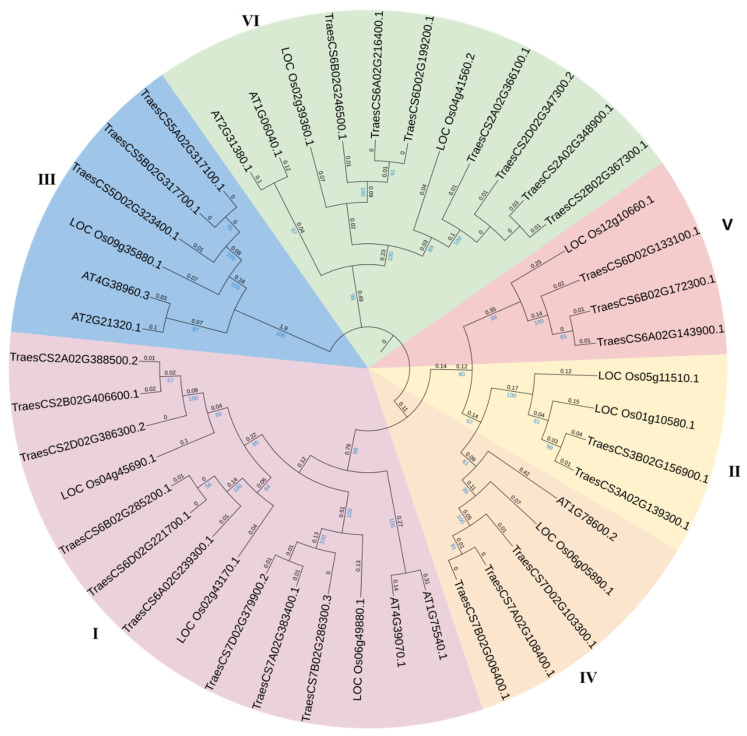
Phylogenetic tree of DBB protein from wheat, *Arabidopsis thaliana*, and *Oryza sativa*. The protein sequences from three species were aligned using ClustalW, and the unrooted phylogenetic tree was constructed with the maximum-likelihood (ML) method. Bootstrap (Blue) values (≥50%) were shown on branches in the phylogenetic tree.

**Figure 3 ijms-25-11654-f003:**
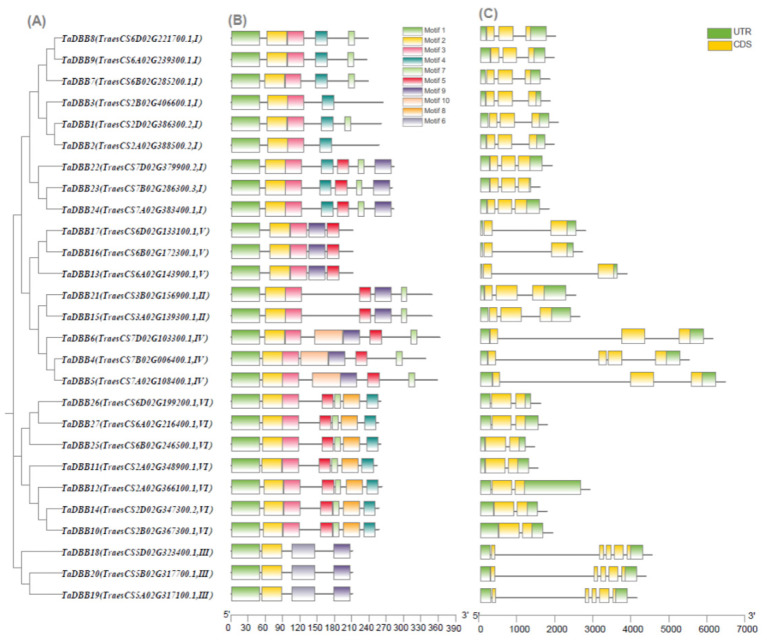
Phylogenetic relationships, motif compositions, and coding gene structures of wheat DBB proteins. (**A**) Multiple alignment of DBBs proteins in wheat. (**B**) Conserved motifs of TaDBB proteins. (**C**) Gene structures of 27 *TaDBB*s.

**Figure 4 ijms-25-11654-f004:**
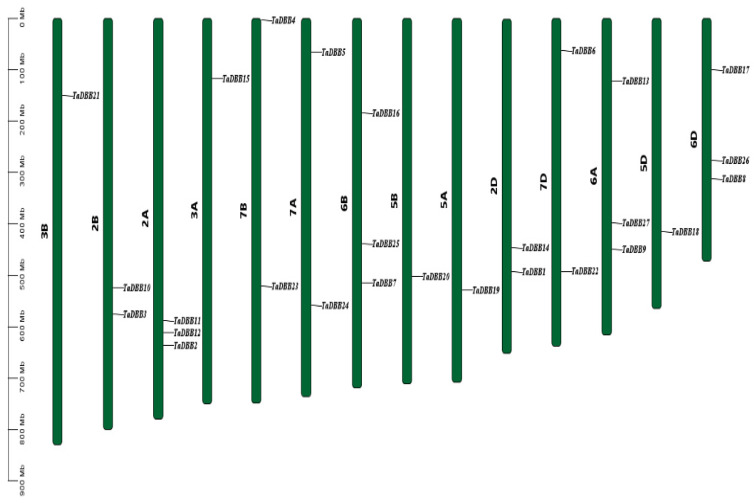
Chromosomal distribution of wheat *DBB* genes. Chromosome locations are referred to in IWGSC RefSeq V1.1 [31].

**Figure 5 ijms-25-11654-f005:**
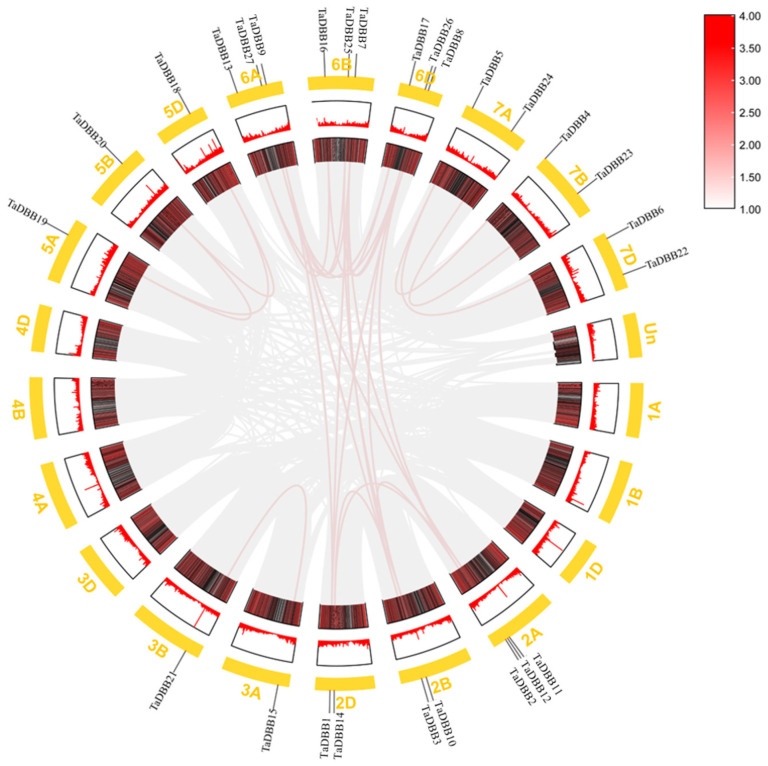
Wheat *DBB* gene family synteny analyses. Syntenic *TaDBB* gene pairs are denoted with red lines, and the other gene pairs in the wheat genome are denoted with gray lines.

**Figure 6 ijms-25-11654-f006:**
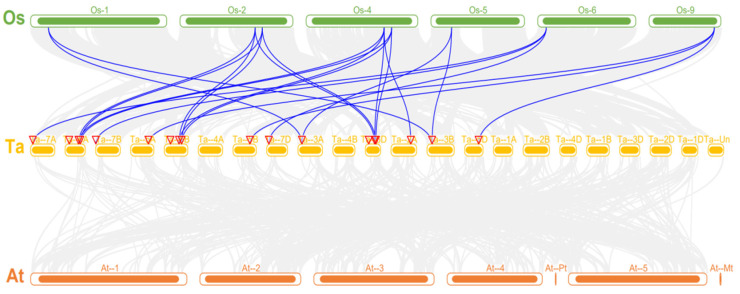
Homology analysis of DBB genes in *Arabidopsis thaliana* (At), rice (Os), and wheat (Ta). Homologous gene pairs in the genomes of wheat, rice, and *Arabidopsis thaliana* are marked with gray lines, while *TaDBB* gene pairs showing homology in the genomes of wheat, rice, and *Arabidopsis thaliana* are marked with blue lines.

**Figure 7 ijms-25-11654-f007:**
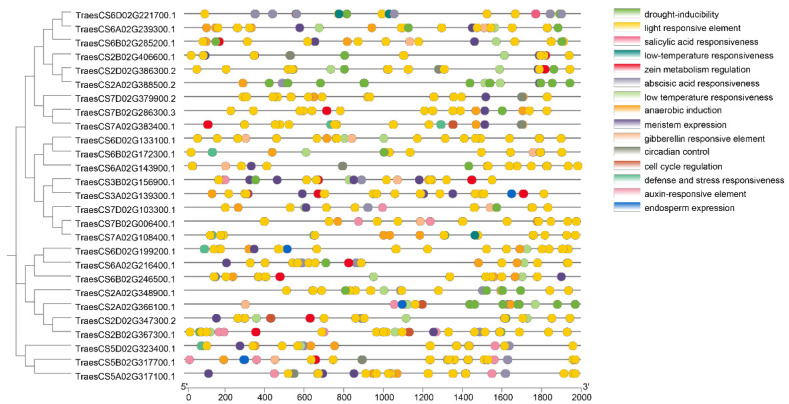
Analysis of cis-elements of 27 TaDBB promoters using the Plantcare database.

**Figure 8 ijms-25-11654-f008:**
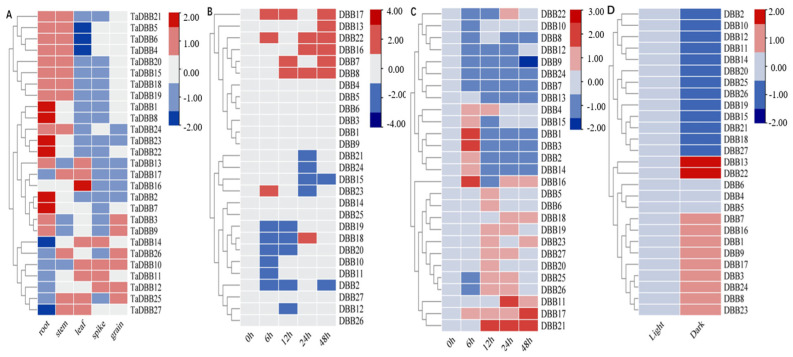
Expression patterns of the 27 *TaDBB* genes. (**A**) Expression level of the *TaDBB* genes in five tissues with data collected from WheatOmics 1.0. The expression of 27 *TaDBB* genes under ABA treatment (**B**), drought stress, (**C**) and light/dark, (**D**) with data collected from qRT-PCR.2.7. Prediction and Validation of Interaction Proteins.

**Figure 9 ijms-25-11654-f009:**
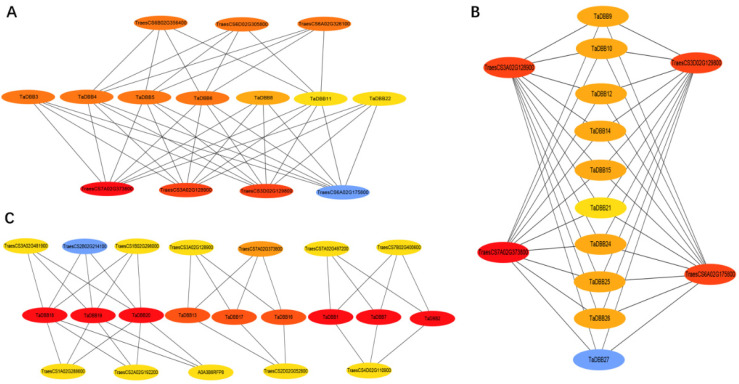
The protein–protein interaction (PPI) network models for members of the TaDBB protein family predicted using the STRING database. (**A**–**C**) showed the interaction network module of different TaDBBs.

**Figure 10 ijms-25-11654-f010:**
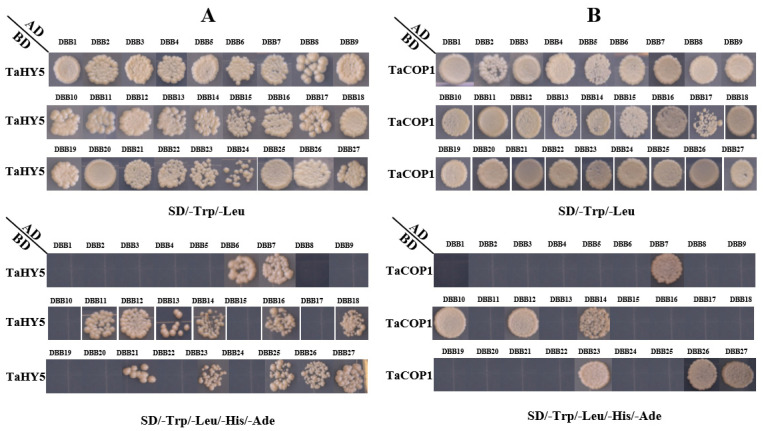
Yeast two-hybrid (Y2H) assays for the 27 TaDBB proteins with TaCOP1 and TaHY5. (**A**) The interaction between TaHY5 and TaDBB1-27. (**B**) The interaction between TaCOP1 and TaDBB1-27.

## Data Availability

The original contributions presented in the study are included in the article/Appendix A; further inquiries can be directed to the corresponding authors.

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
