# Peer review of "Genome-Wide Identification, Characterization and Expression Patterns of the DBB Transcription Factor Family Genes in Wheat"

_ijms, 2024, doi:10.3390/ijms252111654_

Round 1

Reviewer 1 Report

Comments and Suggestions for Authors

To: MDPI IJMS Editorial Office

Manuscript ID: ijms-3264489 - Review Report

Please find my review for the manuscript ID: ijms-3264489 entitled as “Wang et al., Genome-Wide Identification, Characterization and Expression Patterns of the DBB Transcription Factor Family Genes in Wheat

The manuscript is clearly articulated. However, the literatures reviews made for introduction part looks too shallow. There is no relevant information discussed in relation to role of DBB TFs in model and cultivated crops specific to rice, maize, cotton, etc. in terms of abiotic stresses management, such as drought, salt. The biological function of two B-Box domains in TFs family genes functional diversity should elaborated. Research studies made (DBB TFs ectopic expression or knockout/down) and findings obtained specially in crop plants should be surveyed and elaborated in detail to justify the importance TaDBB TFs in wheat.

In general, appropriate data with figures were generated. The results and discussions were also well presented.   

Question-1: Why don’t validate subcellular localization prediction of TaDBB specially for most promising candidate genes by in vivo transient expression assay? Line #78-79

Question-2: What could be the justification of the authors for those TaDBB genes contrasting expression pattern variabilities at given time frame rages? See line #162-169. It needs to be clearly defined with justification in discussion part of the manuscript.

Thank you!

Author Response

Comment 1: However, the literature review in the introduction seems too superficial, without relevant information on the role of DBB TFs in tolerance to abiotic stresses such as drought and salt in typical and cultivated crops such as rice, maize, and cotton.

Reply 1: Thank you for your valuable feedback on the literature review. I agree that it needs to be more in-depth. I expanded the section to include the role of DBB transcription factors in abiotic stress tolerance in major crops such as rice and Arabidopsis. Specifically, I highlighted how Arabidopsis and rice DBB proteins interact through their B-box domains to regulate responses to drought and UV-B stress, and the roles of OsARID3 and OsPURα. Lines 33-40. Lines 58-67

Comment 2: Why not validate the subcellular localization prediction of TaDBB by in vivo transient expression assays, especially for the most promising candidate genes? Lines 78-79

Reply 2: Thank you for pointing out that the candidate genes have not been well identified yet. We will further identify the candidate genes in the future. After further verification, we will perform in vivo transient expression experiments on the candidate genes to verify the subcellular localization prediction.

Comment 3: How do the authors explain the differential expression patterns of these TaDBB genes within a given time frame? See lines 162-169. This needs to be clearly defined and justified in the discussion section of the manuscript.

Reply 3: Thank you for pointing this out, and I agree with this comment. We have explained the explanation of the differential gene expression of TaDBB genes within a given time frame in the discussion section. Lines 267-275.

Reviewer 2 Report

Comments and Suggestions for Authors

The manuscript entitled "Genome-Wide Identification, Characterization and Expression Patterns of the DBB Transcription Factor Family Genes in Wheat" aimed to "identified and classified wheat DBB genes (TaDBBs), followed by phylogenetic analysis. The structures and conserved domains of these genes and their distribution in wheat genome were analysed. The cis-acting elements and expression patterns of the TaDBB family were studied. We also investigated their expression patterns under ABA, light/dark, and drought treatments to reveal their roles in response to environmental stresses. Finally, yeast two-hybrid experiments was conducted to validate the predicted interactions between TaDBBs and COP1 and HY5."

The manuscript lacks substantial novelty in the field of wheat research, although the results may still hold value. However, my primary concern is the insufficient detail in the methods section. The experimental procedures are described without enough specificity, making it difficult to replicate the study or even to fully comprehend how the results were obtained. For example, the authors mention the use of several software tools but fail to provide the specific conditions under which they were used. Also, the description of the stress experiments is minimal, with little information on the number of biological or technical replicates. These are only some examples within the manuscript. As a result, much of the study’s methodology remains unclear, leaving the reader to make assumptions about how certain results were obtained.

The text comes across as rushed, lacking careful attention to detail and clarity. This impression arises from vague descriptions, incomplete explanations, and inconsistencies in language that suggest the content was not thoroughly revised or polished. Consequently, it gives the reader the sense that the writing process may not have been given adequate time and consideration. Even in the SI files, we have the following information: "Supplementary Materials: The following supporting information can be downloaded at" (? where exactly?)

Figures also need to be improved, as well as their legend. For instance, BS values are not seen in Figure 2, nor the legend with the distance. Thus, it is not a phylogenetic tree, but only a figure.

Author Response

Comment 1: However, my primary concern is the insufficient detail in the methods section. The experimental procedures are described without enough specificity, making it difficult to replicate the study or even to fully comprehend how the results were obtained.

Reply 1:

  1. Thank you for your nice suggestions! I have revised the section to provide more specific details about the alignment and phylogenetic analysis. The updated text now reads:“Multiple alignments of the conserved TaDBB protein sequences were performed using the ClustalW tool. The maximum-likelihood (ML) phylogenetic tree was constructed based on the conserved blocks and best-fit model using MEGA 11 software with 1000 bootstrap replications . The phylogenetic tree was visualized using IToL (https://itol.embl.de/upload.cgi).” Line#330-334
  2.  I have refined the explanation of how the gene structure and motifs were visualized. I used the Gene Structure View (Advanced) tool in TBtools to combine the phylogenetic tree, the identified motifs, and the wheat annotation file (GFF3) to better visualize the DBB family members. Line#342-344
  3. I have refined the section on chromosome localization and collinearity analysis. The updated text now specifies the tools in TBtools used for drawing the physical map of DBB family members, calculating gene density, and generating collinearity diagrams. Additionally, I mentioned the version of the Triticum aestivum genome used (IWGSC RefSeq 1.1) and noted that the genome sequences were obtained from the Ensembl Plants database. Line#349-359
  4. I have refined the section on the transcriptome analysis of TaDBBs in different tissues. The revised text clarifies that the Wheat Omics 1.0 database (http://wheatomics.sdau.edu.cn/) was used to download expression data for various wheat tissues, including leaf, stem, root, grain, and spike, in TPM format. Additionally, I specified that expression heatmaps were generated using log2(TPM + 1) values for 27 TaDBB genes and plotted with TBtools. Line#369-373
  5. I improved the section on TaDBB expression profiling and qRT-PCR analysis. Specifically, I detailed that for drought stress treatment, seedlings were withheld from watering from the Feeks1 to Feeks2 stages and subsequently watered at Feeks2 to simulate drought recovery. I also specified that 100 μM ABA was evenly applied to the leaves of seedlings at the Feeks2 stage. Furthermore, I included the systematic collection of leaf samples at various time points and clarified the RNA extraction and cDNA preparation methods, along with the qPCR protocol details. Line#384-393. Line#406-422.
  6. I have refined the section on the prediction and validation of the TaDBBs protein interaction network. The updated text specifies that total RNA was extracted from different parts of Chinese Spring (leaves, seeds, roots, and stems), and cDNA was synthesized using the PrimeScript™ II Kit. I also included details on the nested PCR used to obtain the sequences of TaDBBs, COP1, and HY5, and clarified the transformation of plasmids into Y2HGold. Line#426-440.

Comment 2: Figures also need to be improved, as well as their legend.

Reply 2: Thank you for your valuable feedback. I have updated Figure 2 to include the branch lengths and bootstrap values, as well as revised the legend to provide clear information about the distances represented. These changes ensure that the figure accurately represents a phylogenetic tree. Line#107-110.

Round 2

Reviewer 2 Report

Comments and Suggestions for Authors

The version has improved when compared to the earlier version, but several sections remain hard to follow. Several comments were answered by the authors in this new version. However, this work is mostly theoretical, and the authors should highlight that. For instance, the conditions of how rna-seq data was obtained, the type of material (young, old), the type of sequencing performed and the initial aims of the study are not mentioned. The authors simply wrote that: "The Wheat Omics 1.0 database (http://wheatomics.sdau.edu.cn/) was used to download expression data for the wheat leaf, stem, root, grain, and spike tissues in the transcript per kilobase per million mapped reads (TPM) format, after which expression heatmaps were constructed using log2(TPM + 1) values of 27 TaDBB genes and plotted with TBtools." - I am sorry but this is not enough to understand where these data is coming from and how results were obtained.

The validation of these results were performed by the authors through rt-PCRs using leaves, but again no correlation with the first part of this study was made. Additionally, heat maps are then presented as "Expression patterns of the 27 TaDBB genes. Expression level of the TaDBB genes in five tissues (A), under ABA treatment (B), drought stress (C), light/dark (D). Expression data were presented as the mean data from three biological replicates". Where does this data come from if authors extracted only leaves? And what exactly is being shown in this graph?

Author Response

Comment 1: For instance, the conditions of how rna-seq data was obtained, the type of material (young, old), the type of sequencing performed and the initial aims of the study are not mentioned.

Reply 1: Thank you for your nice suggestions! I have refined the section on the transcriptome analysis of TaDBBs in different tissues. The revised text explains where the data came from and how they were obtained. RNA-seq data of five tissues each at three different developmental stages. Line#371-376 

Comment 2: The validation of these results were performed by the authors through rt-PCRs using leaves, but again no correlation with the first part of this study was made. Additionally, heat maps are then presented as "Expression patterns of the 27 TaDBB genes. Expression level of the TaDBB genes in five tissues (A), under ABA treatment (B), drought stress (C), light/dark (D). Expression data were presented as the mean data from three biological replicates". Where does this data come from if authors extracted only leaves? And what exactly is being shown in this graph?

Reply 2: Thank you for your valuable feedback. For Figure 8, the previous annotation was unclear and has been improved now. The data of Figure 8A comes from the website WheatOmics 1.0 collected from a reported study, and the data of Figure 8B-D comes from qPR-PCR we performed. Figure 8A shows the expression of TaDBB genes in different tissues. Figure 8B-D shows the changes in TaDBB genes expression under different stresses. Line#210-212

Round 3

Reviewer 2 Report

Comments and Suggestions for Authors

I have no further comments.